# A PrEP decision aid for women survivors of intimate partner violence: Task-shifting implementation to domestic violence service settings

**Jaimie P. Meyer**[1,2]*, **Elizabeth Lazarus**[3], **Karlye Phillips**[3], **Z. Thomasina Watts**[3], **Brenice Duroseau**[4,5], **Cindy Carlson**[6], **Carolina R. Price**[1], **Trace Kershaw**[7], **Tiara C. Willie**[3]

1 Department of Medicine (Infectious Diseases), Yale School of Medicine, New Haven, CT, United States of America, 2 Chronic Diseases Epidemiology, Yale School of Public Health, New Haven, CT, United States of America, 3 Department of Mental Health, Johns Hopkins Bloomberg School of Public Health, Baltimore, MD, United States of America, 4 Johns Hopkins University School of Nursing, Baltimore, MD, United States of America, 5 Center for Infectious Disease and Nursing Innovation, Baltimore, MD, United States of America, 6 Umbrella Center for Domestic Violence Services, New Haven, CT, United States of America, 7 Social and Behavioral Sciences, Yale School of Public Health, New Haven, CT, United States of America

* Jaimie.meyer@yale.edu

## Abstract

### Background

Women exposed to intimate partner violence (IPV) experience multiple social and structural barriers to accessing HIV pre-exposure prophylaxis (PrEP), despite being at increased risk for HIV. In addition, few existing HIV prevention interventions address IPV. A recently developed PrEP decision aid for women has the potential to reach IPV survivors at risk for HIV if it could be integrated into existing domestic violence agencies that prioritize trust and rapport with female IPV survivors. Leveraging non-traditional service delivery mechanisms in the community could expand reach to women who are IPV survivors for PrEP.

### Methods

We conducted qualitative interviews and online qualitative surveys with 33 IPV survivors and 9 domestic violence agency staff at two agencies in Connecticut. We applied the Consolidated Framework for Implementation Research (CFIR) to understand barriers and facilitators to delivering a novel PrEP decision aid to IPV survivors in the context of domestic violence service agencies.

### Results

Most IPV survivors and agency staff thought the PrEP decision aid intervention could be compatible with agencies' existing practices, especially if adapted to be trauma-responsive and delivered by trusted counselors and staff members. PrEP conversations could be packaged into already well-developed safety planning and wellness practices. Agency staff noted some concerns about prioritizing urgent safety needs over longer-term preventive

**Data Availability Statement:** All relevant data are within the manuscript and its Supporting Information files.

**Funding:** Support for this project provided by the Center for Interdisciplinary Research on AIDS (CIRA) Pilot Project Awards (to JPM and TCW), which is supported by the National Institute of Mental Health (P30MH062294). The content is solely the responsibility of the authors and does not necessarily represent the official views of the Center for Interdisciplinary Research on AIDS, the National Institute of Mental Health, or the National Institutes of Health. Career development support for TCW provided by the National Institute on Minority Health and Health Disparities (K01MD015005).

**Competing interests:** The authors have declared that no competing interests exist.

health needs during crisis periods and expressed interest in receiving further training on PrEP to provide resources for their clients.

## Conclusions

IPV survivors and agency staff identified key intervention characteristics of a PrEP decision aid and inner setting factors of the service agencies that are compatible. Any HIV prevention intervention in this setting would need to be adapted to be trauma-responsive and staff would need to be equipped with proper training to be successful.

## Introduction

Intimate partner violence (IPV) significantly increases women's risk of acquiring HIV [1]. Globally, women are disproportionately impacted by HIV [2] and women who have experienced physical and sexual IPV are 1.4- and 2-times more likely, respectively, to acquire HIV compared to women who have not experienced IPV [3]. HIV risk among women experiencing IPV is driven by direct exposure to high-risk partners through sex and substance use, and social and structural systems that deter women from accessing effective HIV prevention, testing, and treatment [4]. Despite well-documented associations between HIV and IPV, there remains a shortage of HIV risk reduction interventions that address IPV [5].

For women experiencing IPV, reliance on traditional approaches to HIV prevention, such as male condoms, is often insufficient because condom use requires male participation, and condom negotiation with male partners may precipitate IPV [6]. In contrast, HIV pre-exposure prophylaxis (PrEP) is a user-controlled, event-independent intervention that effectively prevents HIV and is an empowering HIV prevention option for women experiencing IPV [7–12]. Despite clear clinical benefits and women's expressed inclinations to use PrEP once they are aware of it, there continue to be significant barriers to advancing IPV-exposed women along the PrEP care continuum [13–15]. By 2019, fewer than 10% of U.S. women who were clinically eligible for PrEP based on any HIV risk, had received it since FDA-approval in 2012 [16–19].

First steps in the PrEP care continuum involve accurate HIV risk perception, HIV testing (and being "diagnosed" as HIV-negative), and PrEP awareness. These steps are often problematic for women at highest risk of HIV, including women who are IPV survivors, and can be effectively addressed using decision aids [20]. Decision aids offer a patient-centered approach that combines evidence-based medicine and patient preferences. This tool encourages active participation in health options that help patients understand and identify their preferences [21]. Patients are supported in making informed and value-based decisions individually or with their healthcare provider. Decision aids can be distributed by medical providers, community health care workers or other staff [22]. Though there are >150 randomized clinical trials of decision aids and well-developed standardized criteria [23], there were previously only two published HIV-related decision aids available that related to HIV testing and antiretroviral treatment [24]. Meyer and colleagues developed and pilot-tested the first published PrEP decision aid and tailored it to women with substance use disorders in drug treatment settings [20]. In a preference-randomized controlled trial, they found it was highly effective at modifying decisional preference for PrEP and increasing PrEP uptake at 12 months despite no active facilitation strategy for linking eligible and interested participants to PrEP clinical care.

The current study aims to address crucial PrEP implementation challenges for women who are IPV survivors by evaluating how a PrEP decision aid could be adapted and integrated into domestic violence (DV) service settings. We selected these settings because they are important points of access to women who are IPV survivors, and because they are community-based settings that provide trauma-informed care, which prioritizes safety, trust, peer support, collaboration, empowerment, and acknowledges cultural, historical, and gender issues [25,26]. Moreover, trauma responsive services constantly adapt to meet clients' needs. Still, DV service providers often have limited experience delivering health interventions and have been left out of broader HIV prevention initiatives [27]. We applied the Consolidated Framework for Implementation Research (CFIR) to understand multilevel barriers and facilitators to integrating a PrEP decision aid for IPV-exposed women into DV service agencies [28]. Findings have important implications for PrEP scale-up among this key population of women at high-risk for HIV and in this novel non-clinical service setting.

## Methods

### Study setting

The project, known as PrEP WAVE (PrEP for Women who are Violence-exposed), took place in two urban centers in Connecticut. Study concept and design were informed at start-up by a community advisory board, organized by the Yale Center for Interdisciplinary Research on AIDS, that included community service providers and clients. In collaboration with the Connecticut Coalition against Domestic Violence (CCADV), we identified two DV agencies as partners for study implementation. Both agencies provide confidential, trauma-responsive services and care coordination for clients and families experiencing IPV. At the time of study initiation, neither organization had any programming or staff training related to HIV risk reduction or PrEP.

### Study design and participants

The research team participated in onsite staff meetings at each DV service agency to introduce the project and identify strategies for recruitment, with minimal disruption to existing workflows. Between February 25, 2020 and August 24, 2020, we recruited women who are IPV survivors and staff from partnering agencies to participate in qualitative interviews as key stakeholders. Fliers were placed throughout each agency (e.g., front desk, lobby) so potential participants could self-refer. Trained research assistants were onsite at each agency 1–2 days per week during the recruitment phase to inform potential participants about the project. Agency staff could refer potential participants by providing basic contact information and preferred method of contact through a HIPAA-secure Qualtrics link and private protected phone line. Agency staff were recruited through direct outreach (e.g., in staff meetings) and internal email distribution lists.

Referred IPV survivors were screened by a trained research assistant in person or by phone for the following inclusion criteria: 1) 18 years or older; 2) self-identified gender as female; 3) reported at least one physical, sexual, or psychological IPV victimization experience by a male partner in the past 6 months; 4) self-reported HIV-negative status; and 5) English- or Spanish-speaking. Inclusion criteria for agency staff were: 1) 18 years or older; 2) currently providing services at a partnering DV agency; and 3) English- or Spanish-speaking.

After enrollment, participants completed an individual interview with a trained research assistant in a private setting at the DV agency using a semi-structured qualitative interview guide. The interview guide for IPV survivors addressed factors contributing to decisional conflict; knowledge, values, and support/resources; HIV prevention needs; PrEP awareness; and

perceived role of IPV in HIV risk. The interview guide for agency staff addressed decisional conflict and constructs within 5 CFIR domains (Intervention Characteristics; Outer Setting; Inner Setting; Characteristics of Individuals; and Process) to inform adaptation and integration of the decision aid into DV agencies [28]. Interviews were audio-recorded then transcribed by research assistants.

Shortly after interviews began, on March 13, 2020, the University placed a hold on all new enrollments in clinical research due to the then-emerging COVID-19 pandemic. DV agencies also ceased in-person activities. All previously referred participants completed interviews by phone. Thereafter, we adapted the semi-structured interview guide into a structured survey in Qualtrics, allowing for free-text answers. The survey also included an eligibility screener (applying the same inclusion and exclusion criteria as above) and an option to indicate anonymous consent. The Qualtrics link was provided to staff via internal e-mail distribution and virtual staff meetings and was disseminated to IPV survivors using each of the agencies' private Facebook pages. Qualtrics surveys contained no identifiable information and participants were each assigned a unique study ID.

IPV survivors who completed qualitative interviews or online surveys received $25 gift cards for their time. Agency staff were not provided monetary compensation per agency regulations.

All eligible participants were invited to participate and provided verbal consent for qualitative interviews; written informed consent was waived. Online survey participants provided brief written consent but could sign anonymously. All procedures were approved by the Yale University Human Investigations Committee (IRB).

## Analysis

**Procedure.** In-depth qualitative interviews were coded in Dedoose by at least three independent reviewers to identify CFIR constructs, using deductive reasoning [28]. We defined the organization as the DV agency, the intervention as PrEP, and the implementation strategy as the decision aid. We applied standardized definitions of CFIR constructs [28]: *Intervention characteristics* relate to the attributes of the intervention that impact the effectiveness of implementation (including intervention source, evidence strength and quality, relative advantage, adaptability, trialability, complexity, design quality and packaging, and cost.) *Outer setting* describes the external effects on implementation (including cosmopolitanism, external policies and incentives, patient needs and resources, and peer pressure.) *Inner setting* relates to the features of the organization (including structural characteristics, networks and communications, culture, implementation climate, and readiness for implementation.) *Characteristics of individuals* describe personal qualities that may influence implementation, such as knowledge and beliefs about the intervention, self-efficacy, individual stage of change, individual identification with organization, and other personal traits. *Process* relates to various stages of implementing the intervention within the organization such as planning, engaging, executing, and reflecting and evaluating [28]. Transcripts were assessed for decisional conflict related to PrEP and HIV risk reduction.

We used four criteria of trustworthiness during our analysis: (1) credibility, (2) transferability, (3) dependability, and (4) confirmability [29,30]. To increase dependability and credibility, we read each transcript multiple times and held several consensus meetings while coding (i.e., prolonged engagement in the data). We also reviewed data from transcripts and surveys to assess similar constructs (i.e., managed data triangulation). To increase transferability of our findings, this study used purposive sampling to focus on key informants. In the Methods sections, we provide a thorough description of the research process to increase confirmability.

**Interpretation.**   Team members discussed these themes and reached consensus on discrepancies. Two team members then independently reviewed the transcripts again to ensure all CFIR constructs were applied consistently and to achieve thematic saturation. Illustrative excerpts were extracted, organized by CFIR constructs and domains. We did not identify speakers by key demographic characteristics because the organizations are small, and this detail may be identifiable.

Online survey data (free text responses to open-ended questions) was imported into Dedoose and similarly coded for CFIR constructs with application of standard definitions and extraction of illustrative excerpts. Sample characteristics were disaggregated by data source (in-depth individual interview vs. online survey) but otherwise, we considered all qualitative data for analysis.

## Results

### Sample characteristics

In total, nine participants (N = three IPV survivors; N = six DV agency staff) completed in-depth qualitative interviews. The mean age of participants was 37 years old (SD = 9.43), and all identified as cis-gender women; half reported racial identity as white.

An additional 33 unique participants completed online surveys (N = 30 IPV survivors; N = 3 agency staff). Mean age of survey participants was 35.1 years old (SD = 5.1 years) among IPV survivors and 35.8 years old (SD = 10.3) among agency staff. Nearly all (98%) IPV survivor participants reported their sex assigned at birth as female, and all study participants (IPV survivors and agency staff) reported gender was female. Almost half (46.7%) of the IPV survivor participants were either married or in a domestic partnership.

As shown in Table 1, in interviews and online surveys, IPV survivors and agency staff illustrated significant challenges and potential facilitators to implementing PrEP interventions in DV service settings, as organized by CFIR domain.

### Intervention characteristics

**Intervention source.**   Most agency staff expressed advantages to their addressing PrEP with their clients, over other service or clinical care providers in the community. They highlighted their established relationships and the importance of trust with survivors that could aid in intervention uptake.

IPV survivors also often identified DV advocates and agency counselors as trusted sources of information to help guide healthcare decisions.

While there was consensus among agency staff about discussing PrEP with clients, some staff and IPV survivors preferred that PrEP education come from other sources. Some staff members explained, "I do not address HIV prevention with my patients because I am typically in a large group setting"; and "…the closest we get to addressing [HIV prevention] is by discussing reproductive coercion/sexual abuse as tactics used by people who cause harm to gain power and control." One staff member said simply, "We typically just do not discuss their sex life." Other IPV survivors felt their healthcare provider would not be the best person with whom to discuss HIV risk or PrEP ("I think a physician can be intimidating.")

**Relative advantage.**   Agency staff perceived PrEP as a valuable HIV prevention option for survivors compared to barrier methods (such as condoms). Staff noted that condoms are problematic for clients because they are an indiscrete form of pregnancy and HIV prevention.

IPV survivors described the pros and cons of various HIV prevention options which they identified as *abstinence, male condoms, female condoms, mutual monogamy,* and *not sharing needles.* No IPV survivors identified PrEP as a possible HIV prevention option.

**Table 1.** Findings by CFIR domain.

| CFIR Domain | Illustrative Excerpt (Source) |
|---|---|
| **Intervention Characteristics** | |
| **Intervention Source** | *Well, I definitely think coming from the advocate can help because for the most part we build a trusting relationship. (DV agency staff)*<br>*[With] a counselor you can usually have a level of trust. The counselor can tell you things that you can trust more that's gonna happen than somebody else telling you. I don't think putting it out there for people to choose a woman that's victimized, she's not gonna get up and all of a sudden go and get tested and get this and get that done and change her life and stuff like that. When somebody breaks your spirit to the point where you're beaten, you're raped, you have no strength for that. That's why with counseling, people will—[if] it's a counselor talking to the person I think will be better. (IPV survivor)*<br>*I feel like it would be more beneficial for it to be on the healthcare provider side. Just from my experience, I don't think we have enough knowledge around it. (DV agency staff)*<br>*I would definitely talk about [HIV prevention] with my gynecologist. . .or PCP. . .I certainly wouldn't want to put myself at risk. And if there was something to prevent something, yeah I think I would. (IPV survivor)* |
| **Relative Advantage** | *I feel like a condom isn't necessarily the best option for people. I don't know of any other preventive—like STD prevention things other than the PrEP that I'm learning about. And—I think that's awesome. It seems like there is really not too many side effects and they could—hopefully take it discretely. (DV agency staff)* |
| **Adaptability** | *Probably during intake. We ask some general health questions, sexual assault questions. . . So, just—I mean we have so many questions already. I don't see any harm in adding one more just saying something like, "Do you think you be at risk for any sexually transmitted diseases?" And then if they say yes, then we could kinda use that as a talking point for PrEP. (DV agency staff)*<br>*I don't know how that question would look on our intake. Just sort of asking it. . .so maybe figuring out how we should go about approaching the topic and what would we even talk about. Would we say, "Are you concerned about HIV?" and "If you are—what are some of your concerns?" and "Do you need any resources?" or "How to access testing sites?" or "Have you thought about PrEP?" (DV agency staff)*<br>*Safety planning with them around everything, like when I call a client, we call off block numbers. So even if that's something that you guys can adopt when you're calling. . .someone who's on PrEP who you know is a domestic violence victim or whatever. Blocking the number when you call so that the abuser can't pick up and call the clinic back. (DV agency staff)* |
| **Complexity** | *I'd say mostly pressure from the partner and then also shame. Again, not feeling comfortable going forward and saying, "Hey I think I might have this STD." (DV agency staff)*<br>*Just again the control thing. Having somebody digging through your purse, going through all your personal items and the risk of them—not even necessarily find the medication, but even finding paperwork or something. (DV agency staff)*<br>*I'm gonna say this very. . . the Black community. . . does not really talk about those things. It's more like. . .hush hush. (DV agency staff)* |
| **Cost** | *I definitely think health and wellness is a part of self-sufficiency. The problem is that we don't get any funding for it. We only get funding for the financial stuff because the banks fund us and the banks. . .don't care about that stuff. (DV agency staff)*<br>*So definitely the money because they don't have money for condoms. I mean birth control half the time. How do they get to the appointment? For me, yes, I have a car, but it's a pain for me to pay for the gas and the parking to get there. Sometimes the parking in the [organization] is $3 an hour. I can't afford that. You want to prevent pregnancy, but you can't even afford to go park. (IPV survivor)*<br>*First off, if they share the same insurance, and they need to go somewhere for services or medication or something that has to go through insurance, their abusers might very much so be tracking that and monitoring that. (DV agency staff)*<br>*I don't know how insurance and stuff works for PrEP. . . Because if the person's under their partner's insurance, there's the potential of them finding out they're taking this medication that way. (DV agency staff)* |

*(Continued)*

**Table 1.** (Continued)

| CFIR Domain | Illustrative Excerpt (Source) |
|---|---|
| **Outer Setting Domain** | |
| **External Policy** | *It seems that there is a lot of advancement, but the information isn't being pushed out. (DV agency staff)*<br>*That's what I'm sayin'. If nobody is giving you the resources to know what this PrEP program is, you're not gonna know. Unless you're advertising—this is big. (DV agency staff)* |
| **Cosmopolitanism** | *Oh, [redacted name] comes in so they deal with sexual assault. So if anything of that nature they can talk to them more in detail… (DV agency staff)* |
| **Inner Setting Domain** | |
| **Compatibility** | *I think it fits well, especially now…that…we have multiple providers coming here. So it's giving people the opportunity to come and get all their services or all their needs met in one sitting rather than having to travel everywhere so… I think you'll get more people that way. And then also if we do incorporate some type of screening question in our intake, which I don't see that as being an issue at all, it would allow us to have that conversation with people and get more people interested. (DV agency staff)* |
| **Relative Priority** | *But before with DV and all of that, it really depended on the situation because my main focus is what are the needs right now. And if not having a roof and food is, then I can talk to them about [PrEP], but they're not going to care. So, as long as I can get them into a stable place, then we can start talking about the next steps and how to continue…But it's not as often for us treat someone that's already in a good stable place and to be able to focus on their own health first. (DV agency staff)*<br>*With DV cases, we don't usually bring it [sexual health] up unless the client brings it up. It's not something that we readily talk about because it's more—with DV, we're more focused on physical safety and that kind of stuff. And we're not dealing with the medical stuff until way later down the line, if at all. It's usually not a huge concern right away… We're focused on crisis mode and what happens after. We're not as focused on preparation and before. (DV agency staff)* |
| **Characteristics of Individuals Domain** | |
| **Knowledge & Beliefs about the Intervention** | *Yeah, I didn't consider it 'cause right now I'm not sleeping around at all. I don't feel like I need it. But at one point I did and I wasn't taking it but… (IPV survivor)*<br>*No, because I don't wanna have ever sex again in my life. I don't wanna be with anybody ever again. That doesn't come into my mind. I have a lot of—I used to have a lot of gay friends, they would talk about HIV quite a bit, but I'm not worried about it because I'm not gonna be sexually active. (IPV survivor)*<br>*I don't know too much—I don't know about the pill—I never—I heard about it. (DV agency staff)*<br>*I think—I mean, everybody's different. So, my body might be immune to it [HIV], your body might not be immune to it. So, [PrEP] might work, then it might not work for the person. You know? You see I took the pill, but then I still got HIV or—you know what I mean? I don't think, with medication like that is always has its ups and downs I feel. I feel like a pill is never always a guarantee that something's gonna prevent you from having HIV. (DV agency staff)*<br>*I think I would like to know about long-term effects. Sometimes you take something for a long time and you find out you—it's gonna twist your fingers. I don't know, I guess that's the type of thing you want to know—what kinds of side effects you're gonna have. (IPV survivor)*<br>*I think I heard of it, I'm not going to lie but I could be mixing it up, but I think it's that HIV protection. Some people take it. I don't know if it's daily or monthly… (IPV survivor)* |
| **Process Domain** | |
| **Planning** | *Yeah I definitely think it's a good idea to work with the agencies around how to include a discussion about medical wellness and safety into a safety plan and how to include PrEP as a piece of that. If it's applicable…Obviously you're not going to force it on people, but as part of a medical safety plan. (DV agency staff)* |

(*Continued*)

**Table 1.** (Continued)

| CFIR Domain | Illustrative Excerpt (Source) |
|---|---|
| Engaging | *Yeah. So, trying to get a couple different providers that offer different things. We have different providers coming in so that way I think people are more likely to gather information if they're walking around, like some type of health fair or resource fair. Cuz I've set up before just domestic violence information and people don't wanna approach the table and then when I'm at health fairs, people come "Oh, let me grab this, let me grab that." (DV agency staff)*<br><br>*I think a seminar. Having them come out to the seminar and be able to see what the—what PrEP is about, the program and get information. I mean, they need to know about the program to understand. I mean we can advertise and say you know we happen to work with a partner. . . and you know you guys can go talk to them and get information about it and stuff like that. (DV agency staff)* |

**Adaptability.**   Several agency staff members suggested that PrEP counseling, including use of a PrEP decision aid, could be easily adapted into the DV agency intake process and other one-on-one sessions with survivors. They highlighted the ease of incorporating this conversation into discussions they already have with women. Another staff member identified challenges to introducing PrEP during intake. Agency staff highlighted the importance of prioritizing the urgent needs of clients and described safety planning as part of sexual health promotion with survivors. They emphasized different strategies to address safety and privacy when offering any resource that involves PrEP.

## Complexity

Agency staff and IPV survivors highlighted the complexity of implementing HIV prevention in the setting of a DV service agency. Staff described barriers to women making decisions about HIV prevention such as stigma, shame, judgement, and privacy concerns. Another complexity to developing PrEP interventions for IPV survivors is consideration of different sociocultural differences in what is acceptable.

**Cost.**   Some DV agency staff expressed concerns about insufficient funding to implement additional sexual health and prevention initiatives, like a PrEP decision aid, within the agency.

IPV survivors were concerned about the cost of healthcare related to HIV prevention services and issues related to insurance. Multiple agency staff members identified shared health insurance with a partner as an important potential barrier to women accessing PrEP.

## Outer setting domain

**External policy.**   Agency staff highlighted the lack of knowledge and education surrounding PrEP in the community and made recommendations for increasing wider communication about and awareness of PrEP.

**Patient needs & resources.**   In online surveys, agency staff described some key resources that would be helpful to support them providing PrEP information. For example, "it would be great to have a fact sheet, curriculum/workshop, or brochure that discusses resources;" "more fliers and information to have to give my clients;" "printed literature in English, Spanish, and Portuguese." Other staff wanted resources for HIV testing: "I wish I could get trained on the antibody tests;" and "schedule of regular HIV testing dates." Some staff expressed a need for "a contact place and person to refer clients" for HIV testing and linkages to PrEP clinical care.

**Cosmopolitanism.**   One staff member highlighted an external organization that networked with the agency to provide sexual assault services to survivors. Other staff described how the agency's connections with community partners helped provide direct linkages to care

(e.g., having the name of a specific healthcare provider), which was particularly important to their Latinx clients.

### Inner setting domain

**Compatibility.**    DV agency staff expressed that a PrEP decision aid would fit well within the culture of the organization and the services that are already provided to survivors.

**Relative priority.**    Agency staff described HIV prevention as a less urgent priority than more critical concerns, such as ensuring physical safety for their clients.

### Characteristics of individuals domain

**Knowledge & beliefs about the intervention.**    Despite concern about HIV risk at times, most IPV survivors interviewed did not consider PrEP to be personally relevant. Other IPV survivors described serious concerns about their personal HIV risk in the online survey, for example, "Recently, my partner is always staying up at night and I'm worried that he will bring HIV back home. I feel like it's risky."

Some staff expressed unfamiliarity with PrEP and HIV, more generally. Survivors expressed lack of familiarity with PrEP, but they also conveyed a desire to learn more information.

### Process domain

**Planning.**    One staff member described how agencies could incorporate PrEP into existing practices as part of a medical safety plan.

**Engaging.**    Staff members recommended various information-sharing events to aid the implementation process and provide PrEP education to survivors.

## Discussion

In qualitative interviews and online surveys with IPV survivors and DV agency staff, we considered potential strategies to support successful integration of a PrEP decision aid into IPV service agencies, which is a novel approach to bringing HIV prevention out of traditional clinical settings and into community-oriented service settings. In delivering a PrEP intervention directly to IPV survivors in a place where they already receive services, there is the potential of reaching a population of women who stand to benefit from PrEP but otherwise lack access. By applying CFIR in analyzing these interviews and surveys [28], we were able to disentangle some important issues related to PrEP-related intervention delivery to IPV survivors within the context of DV service agencies.

Agency staff and clients both identified the importance of building trust and rapport to address sexual health and HIV risk openly and honestly. Some staff felt they would be well-suited to have these personal conversations, while others expressed concern about their lack of expertise and comfort in this area, saying healthcare providers would be better prepared to address sexual health with agency clients. On the other hand, some women expressed distrust in healthcare providers, which may limit engagement in PrEP. Prior research has described IPV-related medical mistrust and how it differs by IPV exposure type [27,31,32]. To build trust within a DV setting, it is recommended that providers prioritize survivor needs and assure that services are readily available, equitably distributed, and consistently available. Trustworthiness is also developed when providers are effective and culturally competent, as well as positively oriented towards survivors (Kennedy, 2023). These provider characteristics are supported when providers are well trained and experiencing job satisfaction and should be evaluated rigorously by organizational leadership.

A second major challenge to implementation of PrEP-related interventions in DV service agency settings relates to prioritization. When IPV survivors present to the agency in crisis, staff must put immediate basic subsistence needs and safety first above all else. Once clients stabilize, staff may be better able to address other health and social needs. One problem, then, is that agencies and their clients may be de-prioritizing linkages to PrEP at the time when clients are experiencing highest risk for HIV. Our findings suggest that the groundwork for PrEP linkages to care (including interventions to raise PrEP awareness and align personal HIV risk perception) must precede these moments of crisis. Infrastructure needs to be in place to initiate PrEP as easily as possible, for example by providing onsite HIV testing and counseling. Linkages to PrEP services may also be expedited when there are established collaborations with community healthcare providers- linkages that many DV agencies already have in place and use for post-exposure prophylaxis (PEP) for sexual assault survivors. Moreover, DV agency staff need to be equipped with basic training on HIV prevention, sexual health, and PrEP to be able to support their clients.

Decision aids address these issues of PrEP awareness and HIV risk perception and could be routinely delivered within the agency outside of crisis intervention periods, for example during patient intake procedures and during routine follow-up visits. It is not yet known how to best integrate PrEP decision aids into DV agencies and what impact decision aid delivery within DV agencies will ultimately have on women's PrEP initiation. We are currently in the process of conducting a clinical trial of different implementation strategies, including individual- and shared- decision aid completion with DV agency staff following organization-wide staff training (NCT05614492). Findings from this study are forthcoming.

According to DV agency staff, PrEP may not seem personally relevant to IPV survivors because many IPV survivors do not perceive themselves as at-risk for HIV. Instead of framing PrEP as standalone HIV prevention, PrEP could be reframed as a part of safety planning or overall wellness- areas in which IPV agencies are already well-versed and have capacity for scale-up. When PrEP is presented as an empowerment tool to build women's self-sufficiency and independence, it makes more sense in the context of DV service agencies. Acknowledging the stigmatism and shame that communities carry regarding HIV and IPV is an important step in designing the infrastructure for interventions. Wellness reframing empowers people with factual information that can move past these social and cultural barriers to PrEP.

Our interviews with key stakeholders emphasize that PrEP interventions must be tailored to be trauma responsive, as evidenced by recurrent themes of trust, rapport, choice, and collaboration. PrEP interventions for IPV survivors must address potential disclosure of PrEP to partners, which may precipitate IPV or distress within the relationship. IPV survivors may be concerned about how visits with a medical provider or prescriptions for PrEP appear on Explanation of Benefits forms, which is especially salient if their abusive partner is the primary insurance holder. Some states do have in place confidentiality provisions for insured dependents if healthcare providers are aware or survivors know to ask for it [33,34]. PrEP navigators can be extremely helpful with guiding patients through the process of requesting confidentiality as an insured dependent. Survivors also may be wary of bringing home a pill bottle for PrEP that their partner could discover. Long-acting injectable PrEP (e.g., cabotegravir) provides one workaround to this problem because it can be given once every 2 months in a doctor's office and was recently FDA-approved for cis-gender women. This obviates the need for take-home pill bottles that could inadvertently disclose that the patient is taking PrEP.

Study findings have important implications for delivery of PrEP interventions to IPV survivors in the context of DV service settings, but the study was not without limitations. The study took place in two service agencies in major urban centers in Connecticut. Participants were predominately white women; findings may be less transferable to service agencies in other

regions where the sociocultural profile of IPV survivors differs. The agencies are accessed by people from a broader range of sociocultural backgrounds, but there may be a reluctance by some people to participate in this research that is described as personal and difficult to contemplate or culturally unaccepted. Due to pandemic-related restrictions, assessments were not delivered in the same way for all participants, with some participating in in-depth qualitative interviews in person, others in in-depth qualitative interviews by phone, and others via an online survey, though all procedures were informed by agency leadership to generate minimal disruption in workflow. These different data sources may have different degrees of validity and reliability, and participants who completed the online surveys were slightly older in this relatively small sample, but we considered all data as reflective of people's experiences because there was otherwise no evidence-based way to weigh the data.

## Conclusions

IPV survivors experience elevated HIV risk and could benefit from PrEP but are often unable to access it because of misperceptions of personal risk, stigma around disclosing HIV risk or IPV to healthcare providers, anticipated stigma of PrEP, and concerns that PrEP disclosure to partners may exacerbate violence in the relationship. Implementing a PrEP decision aid within an DV service agency, where staff prioritize safety and trust building, is feasible and can be delivered in a trauma-responsive way. Future research should focus on how to best integrate conversations about PrEP for IPV survivors into the normal workflow of DV agencies, offering the opportunity to de-stigmatize and normalize PrEP, while simultaneously increasing PrEP awareness.

## Supporting information

**S1 File. PLoS human participants research checklist.**
(DOCX)

**S2 File. Yale human research protection program correspondence.**
(DOCX)

## Acknowledgments

We wish to thank the study participants, who have entrusted us with their stories. This work was made possible by partnerships with the Connecticut Coalition against Domestic Violence, and leadership at collaborating DV service agencies, the Center for Family Justice and the Umbrella Center for Domestic Violence services (with special thanks to Deb Greenwood, Amanda Posila, Kayte Cwikla-Masas, and Ashley Starr Frechette).

## Author Contributions

**Conceptualization:** Jaimie P. Meyer, Tiara C. Willie.

**Data curation:** Jaimie P. Meyer, Tiara C. Willie.

**Formal analysis:** Jaimie P. Meyer, Elizabeth Lazarus, Karlye Phillips, Z. Thomasina Watts, Carolina R. Price, Tiara C. Willie.

**Funding acquisition:** Jaimie P. Meyer, Tiara C. Willie.

**Investigation:** Jaimie P. Meyer, Carolina R. Price, Tiara C. Willie.

**Methodology:** Jaimie P. Meyer.

**Project administration:** Jaimie P. Meyer, Cindy Carlson, Carolina R. Price, Tiara C. Willie.

**Supervision:** Jaimie P. Meyer, Carolina R. Price, Tiara C. Willie.

**Writing – original draft:** Jaimie P. Meyer, Elizabeth Lazarus, Karlye Phillips, Z. Thomasina Watts, Tiara C. Willie.

**Writing – review & editing:** Jaimie P. Meyer, Elizabeth Lazarus, Karlye Phillips, Z. Thomasina Watts, Brenice Duroseau, Cindy Carlson, Carolina R. Price, Trace Kershaw, Tiara C. Willie.

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
