## [Decision Letter · Decision Letter 0]

3 Jul 2024

PONE-D-23-33363A PrEP Decision Aid for Women Survivors of Intimate Partner Violence: Task-Shifting Implementation to Domestic Violence Service SettingsPLOS ONE

Dear Dr. Meyer,

Thank you for submitting your manuscript to PLOS ONE. After careful consideration, we feel that it has merit but does not fully meet PLOS ONE’s publication criteria as it currently stands. Therefore, we invite you to submit a revised version of the manuscript that addresses the points raised during the review process.

We look forward to receiving your revised manuscript.

Kind regards,

Joel Msafiri Francis, MD, MS, PhD

Academic Editor

PLOS ONE

 [Support for this project provided by the Center for Interdisciplinary Research on AIDS (CIRA) Pilot Project Awards (to JPM and TCW), which is supported by the National Institute of Mental Health (P30MH062294). The content is solely the responsibility of the authors and does not necessarily represent the official views of the Center for Interdisciplinary Research on AIDS, the National Institute of Mental Health, or the National Institutes of Health. Career development support for TCW provided by the National Institute on Minority Health and Health Disparities (K01MD015005).].  

Additional Editor Comments (if provided):

Reviewers' comments:

Reviewer's Responses to Questions

**Comments to the Author**

1. Is the manuscript technically sound, and do the data support the conclusions?

Reviewer #1: Yes

Reviewer #2: Yes

2. Has the statistical analysis been performed appropriately and rigorously? 

Reviewer #1: Yes

Reviewer #2: N/A

3. Have the authors made all data underlying the findings in their manuscript fully available?

Reviewer #1: Yes

Reviewer #2: No

4. Is the manuscript presented in an intelligible fashion and written in standard English?

Reviewer #1: Yes

Reviewer #2: Yes

5. Review Comments to the Author

Reviewer #1: This is an interesting study that will have important implications for practice.

The study procedures, including adjustments made at the beginning of the COVID-19 pandemic, are clearly described. The introduction to extant research on decision aids and CFIR constructs is clear and concise.

The manuscript is well-written, and my comments are very minor.

Based on the findings from survivors and service providers, it makes sense for DV counsellors to provide information about PrEP to clients or have this information available in their agencies so that it can be shared or accessed later in the counselling relationship after immediate safety concerns are addressed. Given that many hospitals/ healthcare providers do not screen for IPV and survivors likely attend these services with a more immediate concern, they may not be informed about PrEP unless they have attended for concerns relating to sexual assault or sexual health. DV agencies informing survivors means that they will be empowered to ask their healthcare provider about it.

Can the authors further explain the potential use of a PrEP decision aid within a DV service setting? It is clear from the article that service providers must be informed and trained on what PrEP is, how to access it, etc. so that they can pass this information on to their clients. Would a decision aid be presented by these staff to their clients?

The points in the discussion, such as that professionals may deprioritize linkages to PrEP at the time when clients are experiencing the highest risk for HIV, are clearly articulated. It would be useful if the authors could expand on these points to inform recommendations (e.g., What sort of training do DV staff need and how can they partner with healthcare providers to get this training? Best avenues for presenting information about PrEP?)

Reviewer #2: Introduction is well organized.

Method

• The definition of CFIR construct could stand alone as procedure in methods section. Please elaborate the analytic approach of deductive seasoning. How does Trustworthiness conduct?

• How does the authors integrate the interview and survey information? The two types of data source may have different nature. Interviews can provide in-depth insights and detailed narratives, while survey text responses can offer broader trends and patterns. How does the authors consider Validity and Reliability of the two different data source?

• What does trauma-responsive service include?

Results:

• The participants completed interview seems older than participants completed online surveys, would there be a real difference. How does the authors deal with these potential differences that influences on the study.

• It is better to show the participants information in the table.

• It would be better to organize the results in a table based on CFIR framework.

Discussion

• The authors mentioned “Our interviews with key stakeholders emphasize that PrEP interventions must be tailored to be trauma responsive ”， But there is no corresponding results for this.

• Authors should provide the future direction.

6. PLOS authors have the option to publish the peer review history of their article (what does this mean?). If published, this will include your full peer review and any attached files.

Reviewer #1: No

Reviewer #2: No

---

## [Author Response · Author response to Decision Letter 0]

24 Aug 2024

August 19, 2024

Joel Msafiri Francis, MD, MS, PhD

Dear Dr. Francise: 

On behalf of our coauthors, thank you for your comments and ongoing consideration of our manuscript, “A PrEP Decision Aid for Women Survivors of Intimate Partner Violence: Task-Shifting Implementation to Domestic Violence Service Settings”. We revised the manuscript based on reviewers’ remarks, and all comments have been addressed as detailed below. All authors have seen and approved the manuscript for submission.

Reviewer Comments Authors’ Response

Please note that funding information should not appear in any section or other areas of your manuscript. We will only publish funding information present in the Funding Statement section of the online submission form. Please remove any funding-related text from the manuscript. Funding information was removed from the manuscript text.

Please include your tables as part of your main manuscript and remove the individual files. Please note that supplementary tables (should remain/ be uploaded) as separate "Supporting Information" files

 The Table was added to the main manuscript.

---

## [Editor Report · Decision Letter 1]

5 Sep 2024

A PrEP Decision Aid for Women Survivors of Intimate Partner Violence: Task-Shifting Implementation to Domestic Violence Service Settings

PONE-D-23-33363R1

Dear Dr. Meyer,

We’re pleased to inform you that your manuscript has been judged scientifically suitable for publication and will be formally accepted for publication once it meets all outstanding technical requirements.

Kind regards,

Joel Msafiri Francis, MD, MS, PhD

Academic Editor

PLOS ONE
---

## [Editor Report · Acceptance letter]

9 Sep 2024

PONE-D-23-33363R1 

PLOS ONE

Dear Dr. Meyer, 

I'm pleased to inform you that your manuscript has been deemed suitable for publication in PLOS ONE. Congratulations! Your manuscript is now being handed over to our production team.

Kind regards, 

on behalf of

Prof. Joel Msafiri Francis 

Academic Editor

PLOS ONE